# Magnons and magnetic fluctuations in atomically thin MnBi$_2$Te$_4$

David Lujan[1,2,11], Jeongheon Choe[1,2,11], Martin Rodriguez-Vega[3✉], Zhipeng Ye[4], Aritz Leonardo [5,6], T. Nathan Nunley[1,2], Liang-Juan Chang[1,7], Shang-Fan Lee[7], Jiaqiang Yan[8], Gregory A. Fiete [9,10], Rui He [4✉] & Xiaoqin Li [1,2✉]

Electron band topology is combined with intrinsic magnetic orders in MnBi$_2$Te$_4$, leading to novel quantum phases. Here we investigate collective spin excitations (i.e. magnons) and spin fluctuations in atomically thin MnBi$_2$Te$_4$ flakes using Raman spectroscopy. In a two-septuple layer with non-trivial topology, magnon characteristics evolve as an external magnetic field tunes the ground state through three ordered phases: antiferromagnet, canted antiferromagnet, and ferromagnet. The Raman selection rules are determined by both the crystal symmetry and magnetic order while the magnon energy is determined by different interaction terms. Using non-interacting spin-wave theory, we extract the spin-wave gap at zero magnetic field, an anisotropy energy, and interlayer exchange in bilayers. We also find magnetic fluctuations increase with reduced thickness, which may contribute to a less robust magnetic order in single layers.

[1] Department of Physics and Center for Complex Quantum Systems, The University of Texas at Austin, Austin, TX 78712, USA. [2] Center for Dynamics and Control of Materials and Texas Materials Institute, The University of Texas at Austin, Austin, TX 78712, USA. [3] Theoretical Division, Los Alamos National Laboratory, Los Alamos, NM 87545, USA. [4] Department of Electrical and Computer Engineering, Texas Tech University, Lubbock, TX 79409, USA. [5] Donostia International Physics Center, Paseo Manuel de Lardizabal 4, 20018 San Sebastian, Spain. [6] EHU Quantum Center, Universidad del País Vasco/ Euskal Herriko Unibertsitatea UPV/EHU, 48940 Leioa, Spain. [7] Institute of Physics, Academia Sinica, Taipei 11529, Taiwan. [8] Materials Science and Technology Division, Oak Ridge National Laboratory, Oak Ridge, TN 37831, USA. [9] Department of Physics, Northeastern University, Boston, MA 02115, USA. [10] Department of Physics, Massachusetts Institute of Technology, Cambridge, MA 02139, USA. [11]These authors contributed equally: David Lujan, Jeongheon Choe. ✉email: rodriguezvega.physics@gmail.com; rui.he@ttu.edu; elaineli@physics.utexas.edu

The emergence of van der Waals (vdW) magnetic materials has provided rich opportunities to explore magnetic orders in systems that continuously approach the true two-dimensional (2D) limit[1–4]. While it is widely recognized that no long-range magnetic order can exist in 2D for an exchange isotropic system[5], real materials need to be taken into account to predict or detect magnetic order in vdW magnets, known to be highly anisotropic. For example, long-range magnetic order is found to persist down to the monolayer limit in $FePS_3$[6] and $CrI_3$[1,7], but breaks down in $NiPS_3$ monolayers[8]. The detection of magnons, collective spin excitations above a magnetically ordered ground state, serves to confirm the presence of long-range magnetic order. Furthermore, by analyzing magneticfield-dependent magnon resonances, critical information on various energetic terms such as exchange and anisotropy can be extracted. The interplay between these energetic terms ultimately determines the stability of magnetic orders in 2D magnetic materials.

$MnBi_2Te_4$ (referred to as MBT below) and related compounds $MnBi_{2n}Te_{3n+1}$ are newly synthesized materials that combine electronic topology with intrinsic magnetic order (i.e., not via dilute magnetic dopants)[9–18]. The quantum anomalous Hall effect and correlated insulating states have been reported in thin flakes, although the nature of the insulating states is still under debate[19–22]. While the electronic topological properties originate from the Bi and Te layers, the long-range spin ordering is attributed to the magnetic $Mn^{2+}$ ion layers. The interplay between magnetism and electronic topology has been shown to be thickness-dependent[23–25]. Although single septuple layer (SL) MBT is topologically trivial, two or more SLs present distinct topological phases according to their magnetic order and number of layers[9]. At low temperature, the Mn moments in each SL are ferromagnetically (FM) ordered while the Mn moments in the adjacent septuple layer exhibit an antiferromagnetic (AFM) order, leading to a vanishing net magnetic moment in samples with an even number of SLs.

## Results

Here we investigate magnons and magnetic fluctuations in MBT in the ultrathin thickness limit using Raman spectroscopy measurements. We focus on a 2-SL partially because it represents the thinnest sample in which MBT exhibits all three possible magnetic orders: AFM, canted-AFM (c-AFM), and FM when an external magnetic field perpendicular to the 2D plane is tuned. We observe a long-wavelength magnon mode at low temperature under both zero and finite external magnetic fields and in all three magnetic phases, revealing the formation of long-range magnetic order in MBT 2-SLs. In the AFM phase, the unusual selection rule and the lack of magnetic field-dependent frequency shift suggest that two-magnon scattering leads to the observed mode. Using spin-wave theory, we extract a spin-wave gap at zero magnetic field ~0.2 meV with an anisotropy energy of ~0.02 meV and interlayer exchange ~−0.14 meV in the bilayer. The energetic terms as well as the magnon selection rules suggest that MBTs are in a different class of vdW magnets from the widely studied $CrI_3$. Magnetic fluctuations in the few-layer samples and a bulk crystal are quantified by analyzing the quasi-elastic scattering (QES) mode and found to increase with decreasing layer thickness.

An MBT crystal consists of seven atomic blocks or SLs in the sequence of Te-Bi-Te-Mn-Te-Bi-Te stacked along the c-axis. Upon cleaving along the (001) surface, the termination layer is typically a Te layer followed by a Bi layer. Mn atoms form a triangular lattice in each SL with an ABC stacking order between the layers, i.e. a Mn atom of the top layer (dark purple in Fig. 1a) is aligned with the center of a triangle in the bottom layer (lighter purple). The magnetic phase diagram for the 2-SL sample as a function of temperature and magnetic field is illustrated in Fig. 1b, which is informed by recent magnetic circular dichroism spectroscopy measurements on few-layerthick MBT[21,26]. At low temperature and small field, MBT exhibits an AFM phase. At zero magnetic field, the Néel temperature is $T_N \approx 17$ K, slightly lower than the bulk value $T_N \approx 24$ K as determined from transport measurements[18–21]. When a magnetic field is applied perpendicular to the 2D plane, the spin-flop transition occurs at $H_{sf} \approx 2$ T and the FM transition at $H_{FM} \approx 6$ T for a 2-SL. We caution that the precise B-field range over which the c-AFM phase exists in ultrathin samples is still under debate[26,27], and the transition field can differ depending on the magnetic defect density[18,28].

In a 2-SL, the magnetic unit cell contains two Mn atoms, leading to two magnon branches. In each of the three phases, symmetry operations and Raman tensors derived from the magnetic point groups are different (more details in SI). Spin wave theory calculations (analogous to DFT calculations for phonons) are then performed. The calculated magnon wavefunctions carry further symmetry constraints and predict the polarization configurations of the Raman signal associated with a magnon mode (more details in SI).

We illustrate the possible magnon modes in each phase and their invariance under unique symmetry operations in Fig. 1c–e. When the magnetic structure is (not) inversion-symmetric, a magnon mode is Raman active (silent). In the simplest case of the FM phase, only the highfrequency branch $\omega_H$ of the two calculated magnon modes is invariant under inversion $\mathcal{I}$ operation, and thus, Raman active (right panel in Fig. 1e). One expects to observe this magnon mode in the ($\sigma_+/\sigma_+$) configuration, i.e., the incident and scattered photons have the same helicity. In the AFM phase, magnons oscillate with different amplitudes in each layer (Fig. 1c). While these modes break inversion symmetry, they are symmetric following the combined operation of inversion, time-reversal, and a two-fold rotation ($\mathcal{ITC}$). One expects to observe these magnons in the cross-circularly polarized channel ($\sigma_+/\sigma_-$). In the c-AFM phase shown in Fig. 1d, the canted moments remain symmetric under the combined operations of a two-fold rotation and time reversal ($\mathcal{CT}$). In this phase, the high-frequency branch is observable in the co-circular polarization configuration. The low-frequency branch has a vanishing frequency at the Brillouin zone center even in the presence of single-ion anisotropy, and thus, is not observable in Raman experiments. These selection rules, in conjunction with the energy scale, guide us in identifying magnon modes in different phases. However, we caution that they are only applicable to one-magnon scattering process.

We first report magneto-Raman measurements of the FM phase on the 1-SL, 2-SL, and 4-SL samples at 11 K with a 6 T B-field (Fig. 2a). The spectra taken with co-circular (red points) and cross-circular (blue points) polarization are compared to identify the symmetry of the modes. In all displayed Raman spectra, we removed a quasi-elastic scattering (QES) background, which will be discussed later in this paper. After this background removal, Raman spectra from the 1-SL sample exhibit little signal in the frequency range from $-3$ cm$^{-1}$ to $-12$ cm$^{-1}$. We only show the anti-Stokes side of each spectrum because our instrument has an artifact in the ultralow frequency Raman line which obscures real signals from samples in the Stokes side. In both the 2- and 4-SL samples, we observe a phonon mode that persists above the magnetic transition temperature. The co-circular selection rule is in good agreement with that of an $A_g$ phonon mode with a crystalline symmetry point group $D_{3d}$. We attribute this phonon to the layer breathing mode, often observed in vdW materials[29].

In the 2-SL, an additional peak with a central frequency ~5.7 cm$^{-1}$ is observed in the co-circular configuration. To

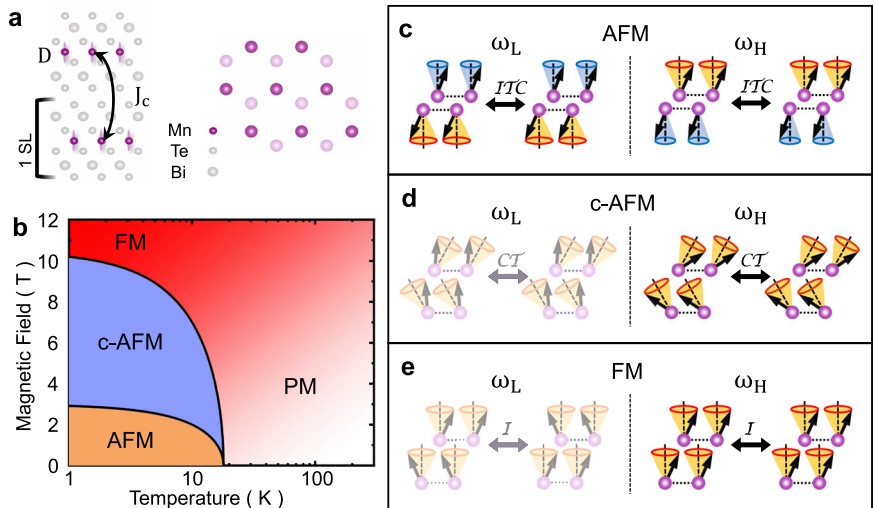

**Fig. 1 Magnetic lattice, phase diagram, and magnon modes in a 2-SL MBT. a** Illustration of two triangular magnetic lattices formed by Mn ions in the top (bottom) layer indicated by dark (light) purple spheres. The layers follow an ABC stacking order. $D$ and $J_c$ represent anisotropy and interlayer exchange coupling. **b** Phase diagram as a function of temperature and an applied magnetic field perpendicular to the 2D plane. **c** Magnons in antiferromagnetic (AFM) phase: spins from two layers oscillate with different amplitudes, corresponding to two magnon modes. Illustrations of spins before and after applying a time-reversal $\mathcal{T}$, inversion $\mathcal{I}$ and two-fold rotational ($\mathcal{C}$ w.r.t $x$-axis) operator ($\mathcal{ITC}$). **d** Magnons in canted AFM (c-AFM) phase: the spin are canted at an angle with respect to the applied magnetic field. Illustrations of spins before and after time-reversal and two-fold rotational ($y$-axis) operator ($\mathcal{CT}$). **e** Magnons in ferromagnetic (FM) phase: spins in the two layers oscillate in-phase or out of phase. Illustrations of spins before and after inversion operator ($\mathcal{I}$). Shaded modes are either Raman silent or near zero frequency. $\omega_L$ ($\omega_H$) refers to the magnon branch with a higher (lower) frequency in the left (right) column.

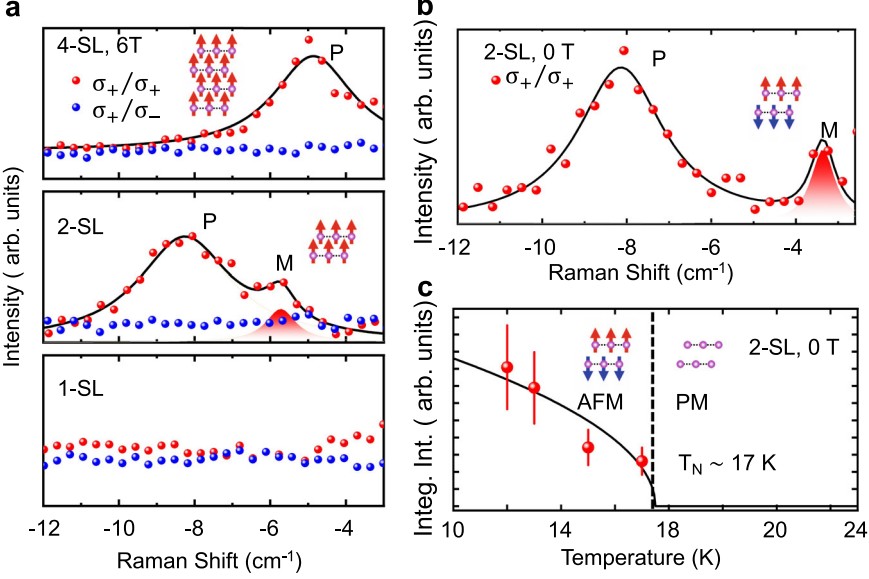

**Fig. 2 Raman spectra of magnon in different setuple layers, magnetic fields, and its temperature dependence at zero field. a** Raman spectra taken at 12 K and 6 T field in 1-SL (bottom), 2-SL (middle), and 4-SL (top) MBT taken with co-circular $\sigma_+/\sigma_+$ (red points) and cross-circular $\sigma_+/\sigma_-$ (blue points) polarized incident and scattered photons. Black solid line is fitting with the sum of two Lorentzian functions, corresponding to a phonon (P) and a magnon (M) mode, respectively. **b** Raman spectrum from the 2-SL taken with co-circular polarizations at 12 K and 0 T. **c** Temperature dependence of the integrated Raman intensity of the ~3.4 cm$^{-1}$ magnon peak from the 2-SL at 0 T. The intensity diminishes as the 2-SL goes through the antiferromagnetic (AFM) to paramagnetic (PM) transition.

elucidate the nature of this mode, we present another Raman spectrum taken at low temperature and zero external B-field, which reveals a mode at ~3.4 cm$^{-1}$ (see Fig. 2b). The intensity of this zero-field mode decreases with temperature and disappears above the Néel temperature of ~17.5 K, suggesting its dependence on the magnetic order. The temperature-dependent intensity can be fitted with a function of $\sqrt{1 - T/T_N}$ (solid line in Fig. 2c),

consistent with the mean-field theory of magnetic moments. The co-circular selection rule, however, contradicts that expected for one-magnon scattering in the AFM phase.

We further study the systematic evolution of Raman spectra from the 2-SL sample at low temperature (12 K) as a function of magnetic field (Fig. 3) in the co-circular polarization configuration. All spectra in Fig. 3a are fitted with two Lorentzian functions

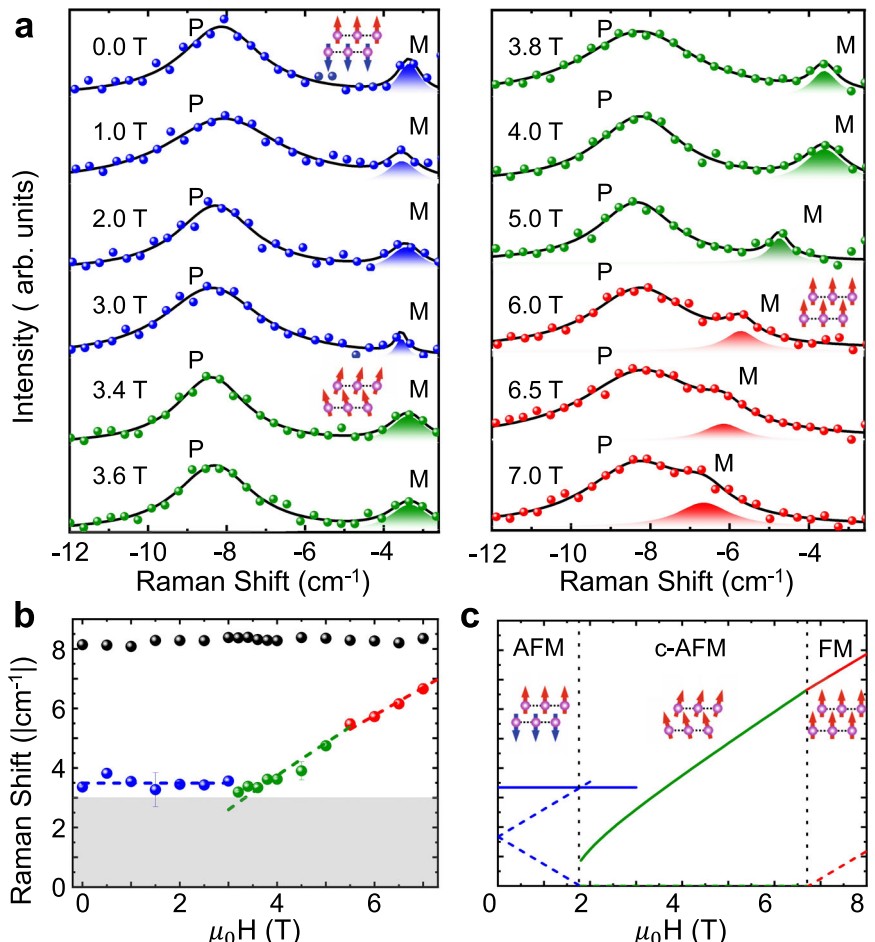

**Fig. 3 Magnetic-field-dependent Raman spectra of the magnon mode from the 2-SL with co-circular $\sigma_+/\sigma_+$ polarization. a** Raman spectra at 12 K as a function of B-field. Black solid line is fitting with the sum of two Lorentzian functions. P and M represent phonon and magnon, respectively. **b** Extracted central frequency for both the phonon and magnon vs. B field. Error bars displayed are smaller than the data points. **c** Calculated magnon modes in the antiferromagnetic (AFM), canted-AFM (c-AFM), and ferromagnetic (FM) phases.

and the extracted central frequencies of the modes are summarized in Fig. 3b. The phonon frequency (black dots) does not show any magnetic field dependence, while the magnon mode shows distinct B-field dependence in different magnetic phases. At the low field, the AFM magnon mode shows a frequency of ~3.4 cm$^{-1}$ independent of the applied magnetic field. Both the magnetic-field independence of the energy and the selection rules (i.e. co-circular polarization) contradict the expectations for a one-magnon process. At 3 T, we observe a discontinuity in the frequency, signaling a transition into the c-AFM phase. The transition between the c-AFM to FM phases at ~5.5 T causes a subtle change in the slope of magnon frequency vs. B-field (more details included in SI). The B fields at which the magnetic phase transitions occur are difficult to identify based on magnon spectra. Our assignments are informed by recent magnetic circular dichroism spectroscopy measurements on few-layer thick MBT[21,26] and calculations presented below.

Theoretically, we model the magnons in 2-SL MBT using a simple spin Hamiltonian with nearest-neighbor interlayer ($J_c$) and intralayer ($J_1$) Heisenberg interactions and single-ion anisotropy ($D$). We obtain the corresponding magnon spectra in each of the three magnetic ground-states within non-interacting spin-wave theory (Fig. 3c). In the AFM and FM phases, the Brillouin-zone center magnons (i.e., at $\Gamma$ point with $k=0$) can be modeled using simple expressions: $\omega_{AFM}^{H(L)}(k=0) = 2\sqrt{SD(SD - SJ_c z_c)} \pm \gamma H_z$, and

$\omega^H(k=0) = 2SD + \gamma H_z$, where $z_c = 3$ is the number of interlayer nearest-neighbors for 2-SL and $\gamma = 0.9343$ cm$^{-1}$/T is the gyromagnetic ratio. The interaction energies $SJ_c$ and $SD$ are used as fitting parameters in our theory, where $S \approx 5/2$. In both cases, we anticipate a simple linear relation between the magnon frequency vs. applied magnetic field $H_z$. The Raman-active magnon in the c-AFM phase follows $\left(\sqrt{SJ_c z_c((\gamma H_z)^2 \frac{SJ_c z_c - SD}{(SJ_c z_c + SD)^2}) + 4SD}\right)$. Fitting the field-dependent magnon frequencies in the cAFM phase, we find magnetic ion anisotropy $SD \approx 0.024 \pm 0.011$ meV and interlayer exchange interaction $|SJ_c| \approx 0.138 \pm 0.013$ meV, with $SD < |SJ_c|$. In Fig. 3c, we plot the predicted magnon modes using the parameters $SJ_c = -0.138$ meV and $SD = 0.024$ meV.

In the AFM phase, the theory predicts two Raman active magnon modes with opposite field-dependent frequency shifts (Zeeman shifts). The calculated AFM one-magnon frequency is below our experimental detection range $\gtrsim 3$ cm$^{-1}$ (shaded area). Summing the energies of these two magnon branches leads to a field-independent two-magnon resonance, close to the observed resonance at ~3.4 cm$^{-1}$. Furthermore, the Loudon–Fleury scattering theory model[30–33] adopted in our calculation predicts that two-magnon scattering is observable in the co-circular polarized configuration, consistent with the experimental observation but in contrast with the selection rule expected for one-magnon scattering (Fig. 1c). The transition between the AFM to c-AFM

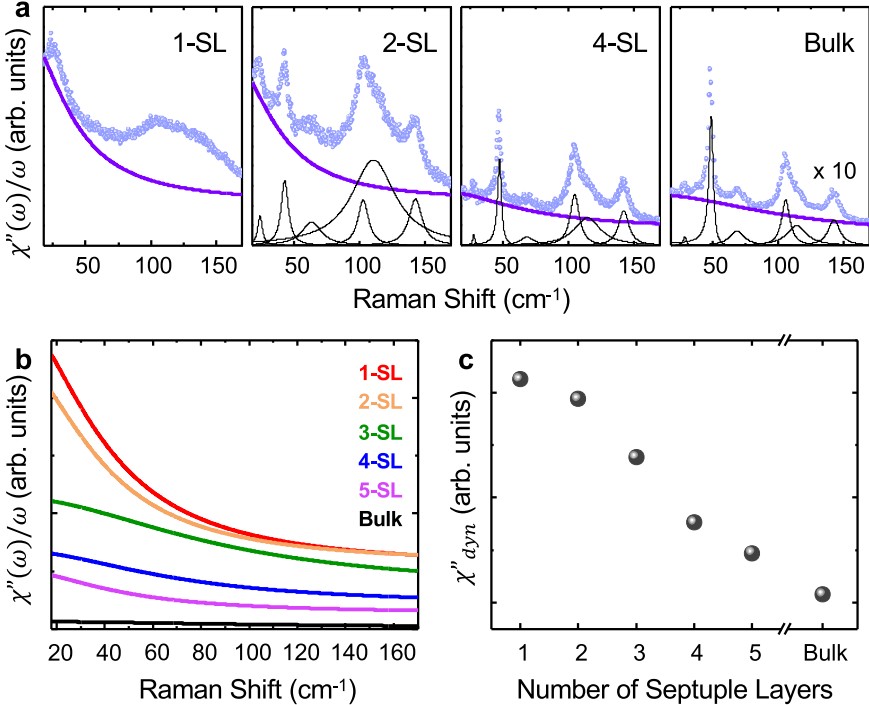

**Fig. 4 Magnetic fluctuations quantified by the QES peak in the paramagnetic phase at 300 K. a** Normalized Raman susceptibility $\chi''/\omega$ for different SLs. The blue dots represent data points. The purple curves are fittings given by Eq. (1). Black solid lines are fitting for several phonon modes using either Lorentzian or Fano functions. **b** Summary of the fitted $\chi''/\omega$ for several flakes with different thicknesses. **c** Layer-dependent dynamic Raman susceptibility $\chi''_{dyn}$ calculated from the integration of $\chi''/\omega$ up to 180 cm$^{-1}$.

phase is predicated to be ~2 T by theory (see Fig. 3c), which is smaller than the 3 T inferred from dichroism spectroscopy measurements performed on similar samples[21]. Two-magnon scattering has been reported in Raman spectra of other AFM materials, e.g., LiMnPO$_4$[34,35]. The narrow linewidth of the two magnon scattering peaks in MBT is somewhat surprising and should be further investigated in future studies. The slightly different slopes for magnon frequency vs. B-field in the c-AFM and FM phases are also captured in our theory and are associated with the finite angle between the applied B-field and the magnetic moments.

In all Raman spectra displayed so far, we have removed a background. This background, plotted over a broader frequency range, is attributed to a QES peak with zero frequency shift from the excitation laser (see Fig. 4). This peak originates from magnetic fluctuations due to coupling to the lattice that is stronger at higher temperatures in the paramagnetic phase[36]. We present room-temperature Raman measurements from several flakes with different thicknesses (a more complete data set is included in SI). Multiple phonons superimposed on the QES peak are identified and discussed in the supplementary. For quantitative analysis, we first apply a Bose factor correction to obtain the normalized Raman susceptibility:

$$\frac{\chi''(\omega)}{\omega} = \frac{I(\omega)}{(n+1)\omega} \propto C_m T \frac{\kappa_m k^2}{\omega^2 + \left(\kappa_m k^2\right)^2}, \qquad (1)$$

where $n$ is the Bose–Einstein factor, and $I(\omega)$ is the Raman intensity. The intensity of the peak is proportional to magnetic specific heat $C_m$. The linewidth is determined by the spin-lattice relaxation time in the hydrodynamic theory of spin waves and characterized by $\kappa_m k^2$, where $\kappa_m$ is the magnetic contribution to thermal conductivity[37–39]. The fitted normalized Raman

susceptibility $\frac{\chi''(\omega)}{\omega}$ for multiple flakes is summarized in Fig. 4b. We also obtain the dynamic Raman susceptibility $\chi''_{dyn}$ (Fig. 4c) via the Kramers–Kronig relation, $\chi''_{dyn} = (2/\pi) \int_0^\Omega \chi''/\omega dx$ and by fitting the measured QES background from 18 cm$^{-1}$ to 180 cm$^{-1}$ (22.3 meV)[40]. Afterward, we extrapolated the fitting down to 0 cm$^{-1}$ and integrated the spectral response, consistent with the common practice in the literature. We observe that $\chi''_{dyn}$ is peaked at the Nèel temperature, ~17 K (data in SI), which is expected for the critical behavior of magnetic phase transition[8,38] and supports our analysis of the QES. The stronger QES signal in thinner flakes is attributed to increased magnetic fluctuations upon approaching the 2D limit[1,2]. Phonon peaks also broaden as the layer thickness decreases,[41] likely related to increased phonon damping due to coupling to spin fluctuations.

Since the study magnons in atomically thin vdW layers is an emergent field, we offer a brief comparison between MBT and CrI$_3$ bilayers. While the ground state in both bilayers are AFM with FM order within each layer, there are important differences between them. The first difference lies in the symmetry. Cr$^{3+}$ are arranged in a honeycomb lattice, where two Cr$^{3+}$ ions per unit cell reside in two sub-lattices within a monolayer. The second difference is the relative energy scale between interlayer exchange coupling and anisotropy. As a weak interlayer exchange system, the selections rule for magnons in CrI$_3$ bilayers can be derived from those in monolayers with the additional consideration of broken inversion symmetry in the AFM phase. In contrast, MBT bilayers exhibit strong exchange coupling between the layers[42]. Thus, each layer cannot be considered separately, and the magnetic point group of the entire 2-SL must be considered.

These energy scales also have important implications for stability of magnetic order in the single-layer limit. From the measured magnon modes in MBT bilayer, we extract a spin wave gap at zero

magnetic field $2\sqrt{SD(SD - SJ_c z_c)} \approx 0.2$ meV using anisotropy $SD = 0.024$ meV and interlayer exchange $SJ_c = -0.138$ meV. Since magnetic anisotropy stabilizes the long-range magnetic order in these 2D magnets[43], the weak magnetic anisotropy energy may contribute to a less robust magnetic order in 1-SL MBT. In contrast, the anisotropy energy in $CrI_3$ is estimated to be 0.27 meV from Raman measurements[7], which is nearly one order of magnitude stronger and sufficient to stabilize the FM order in $CrI_3$ monolayers. This interplay between anisotropy and interlayer exchange also leads to a new c-AFM phase in MBT.

## Discussion

In summary, we report direct observations of magnons in a 2-SL MBT as the ground state is tuned through three magnetically ordered phases by an external field. We were not able to observe a systematic evolution in magnon spectra as a function of layer thickness. In 4-SLs for example, the low-frequency phonon mode spectrally overlaps with and masks the anticipated magnon resonances. Magnetic fluctuations increase with reduced thickness due to enhanced spin-lattice coupling, which may contribute to the absence of magnons in the 1-SL MBT. We cannot completely rule out the possibility that the absence of magnons in 1-SL is related to coupling to the substrate. Future experiments on 1-SLs performed under different conditions (e.g. at higher magnetic fields) are needed. MBT provides an interesting comparison with $CrI_3$ as a strongly coupled exchange system with rich magnetic phases including a c-AFM phase controllable via electric field[44]. A recent scanning tunneling microscopy experiment suggests that electronic structures fluctuate at the atomic scale on the surfaces[45] although it is likely related to disorders that have yet to be better controlled for realizing interesting quantum phases[46]. Our finding complements these prior studies by investigating magnetic orders in ultra-thin MBT layers. Our studies will also guide future investigations of coupled collective excitations and searches for exotic phases. For example, the spectral overlap between a magnon and a phonon may lead to hybridization between these modes, which would fundamentally change the topological nature of magnons in MBT[47,48].

## Methods

**Sample preparation**. High-quality $MnBi_2Te_4$ (MBT) crystals were grown by the flux method[18]. We obtained thin MBT flakes using the $Al_2O_3$-assisted exfoliation method[20,49]. Optical transmittance measurements were used to determine the layer thickness based on the Beer–Lambert law[20]. Ultrathin MBT layers may degrade in an ambient environment. The samples studied here were prepared in a glove box and exposed to air for no more than 1 min. We have also performed measurements on encapsulated samples but found reduced polarization contrast. Thus, we choose to present results from a sample without encapsulation layers. We present low-frequency Raman spectra from 1-SL to 5-SL flakes in the SI. The systematic frequency shift as well as their reasonably narrow linewidths support the high crystalline quality of these ultrathin layers.

**Raman spectroscopy**. Raman measurements were performed using a 632.81 nm excitation laser with a full-width-half maximum (FWHM) of 0.85 cm$^{-1}$. The laser power was kept below ~100 μW to avoid local heating and damage to the samples. The laser beam was focused onto the sample via a ×40 microscope objective to a spot size of about 3 μm in diameter. The Raman signal was collected in the back-scattering geometry and measured with a Horiba LabRAM HR Evolution Raman microscope (1800 grooves/mm grating). All measurements were taken in a closed-cycle helium cryostat from 11 to 300 K with a base pressure lower than $7 \times 10^{-7}$ Torr. An out-of-plane magnetic field ranging from 0 to 7 T was applied.

## Data availability

The data sets generated and/or analyzed during the current study are available from the corresponding author upon reasonable request.

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

## Acknowledgements

We thank Chao Lei, B. Wieder, A. Ernst, and M. G. Vergniory for helpful discussions. This research was primarily supported by the National Science Foundation through the Center for Dynamics and Control of Materials: an NSF MRSEC under Cooperative Agreement No. DMR-1720595, which also supported the facility used in sample preparation. Additional support from NSF DMR-1949701 and DMR-2114825 is gratefully acknowledged by G.A.F. This work was performed in part at the Aspen Center for Physics, which is supported by the National Science Foundation grant PHY-1607611. A.L. acknowledges support from the funding grant: PID2019-105488GB-I00. Z.Y. and R.H. acknowledge support by the NSF CAREER Grant No. DMR-1760668 and NSF Grant No. DMR-2104036. X.L. gratefully acknowledges the Welch Foundation grant F-1662 for support in sample preparation. Work at ORNL was supported by the U.S. Department of Energy, Office of Science, Basic Energy Sciences, Materials Sciences and Engineering Division. M. R-V. was supported by LANL LDRD Program and by the U.S. Department of Energy, Office of Science, Basic Energy Sciences, Materials Sciences and Engineering Division, Condensed Matter Theory Program. L.-J.C. and S.-F.L. were primarily funded by the Ministry of Science and Technology 105-2112-M-001-031-MY3 in Taiwan, and the collaboration with UT-Austin is facilitated by the Air Force Office of Scientific Research under award number FA2386-21-1-4067. Partial funding for L.-J.C. while visiting UT-Austin was provided by a Portugal-UT collaboration grant.

## Author contributions

J.Y. grew the MBT bulk crystals. D.L. prepared the exfoliated samples, and N.N., L.-J.C., and S.-F.L. helped characterize the samples. D.L., J.C., and Z.Y. performed magneto-Raman measurements under the supervision of R.H. and X.L. Data were analyzed by D.L., J.C, R.H., and X.L. The model calculations are led by M.R.V. and G.A.F., and the first-principles calculations are led by A.L.. D.L., M.R.-V., R.H., and X.L. wrote the paper with the input from all authors.

## Competing interests

The authors declare no competing interests.
