## [Peer Review File · Nature Communications]

Reviewers' Comments:

Reviewer #1:

Remarks to the Author:

In the manuscript "Magnons and magnetic fluctuations in atomically thin MnBi₂Te₄" D.Lujan et al. study collective spin excitations (magnons) in ultrathin films of antiferromagnetic topological insulator MnBi₂Te₄ by using Raman spectroscopy technique. They mostly focus on the films composed of 2-seven layer (SL) blocks and study 3 magnetic phases: antiferromagnetic, canted-antiferromagnetic and ferromagnetic ones. After carefully reading the manuscript I clearly see that this is an original paper that contains interesting results on magnetic structure of 2-SL MnBi₂Te₄ system which can be useful for future studies of defect tuning of magnetic properties of MnBi₂Te₄. The paper is clearly written and can be interesting for wide condensed matter community especially in the field of magnetic topological matter.

I can recommend the manuscript for publication in the Nature Communications journal as it is.

Reviewer #2:

None

Reviewer #3:

Remarks to the Author:

In the manuscript of D. Lujan et al., Raman spectroscopy was carried out to probe the magnon of low energy and magnetic fluctuation in few-layer magnetic MnBi₂Te₄. In particular for a 2-SL sample, a signal ~ 5.5 cm⁻¹ was assigned to the photon scatterings with two magnons. Its evolution in perpendicular magnetic fields was used to identify various magnetic phases such as AFM, c-AFM and FM. By contrast, this Raman peak was found to be absent in 1-SL, owing to the strong spin fluctuation.

Different from the more commonly used non-contact techniques that characterize magnetization directly, e.g. MCD and MOKE, Raman is used to probe collective magnetic excitations, such as the magnon and quasielastic scattering presented in the manuscript. Although MBT is being intensively studied by various techniques, there has been few papers about Raman characterization of magnons. In this regard, the manuscript is novel and interesting. Nevertheless, the experimental result is not systematic and conclusive. Several issues need to be clarified.

1. 1-SL has shown to be ferromagnetic at low temperature by MCD and transport measurement. The lack of magnon signal in Raman spectroscopy thus may not be generic. Rather, it could be due to sample degradation.

In addition, for the thicker samples (> 2-SL), the absence of magnetic signal, even in a strong field that would drive the magnon mode away from the low-energy phonon mode, is surprising. The author should clarify the detailed layer dependence, including both odd and even layers.

As such, a major concern is the quality of samples and the repeatability of experimental results. Encapsulation in BN heterostructures is strongly recommended to obtain reliable signals.

2. For the two-magnon mode, any explanation of its narrow width? It is even narrower than the one-magnon mode in high magnetic fields. Again, why is this mode absent in 4-SL? It's not expected to be obscured by the phonon mode ~ 5 cm⁻¹.

3. Due to the variation of sample quality, the spin-flop and FM transition fields are sample dependent. The classification of magnetic phase in Fig. 3 seems not reliable. (a) Check the data of 3 T in Fig. 3a and Fig. S2. From the raw data, one could not distinguish any sudden change. (b) The assignment of 5 T to c-AFM has no evidence, as it could also belong to the FM phase. Overall, the classification should be based on the experimental results in the manuscript rather than sample-dependent characteristics in the literature.

4. Abnormally strong background signals were found for samples of various thickness at room temperature, which were ascribed to QES. (a) Whether is it due to the polymer substrate? The authors need to exclude the fluorescence of organic molecules. (b) It should be suppressed by magnetic ordering below T_N , at least for 2-SL and thicker sample. The authors should show the critical variation around T_N to prove it is indeed related to magnetic fluctuation.

Besides, there are some mistakes to be corrected.

1. In the first paragraph, the author said that "... it breaks down in NiPS3 monolayers and bilayers". This contradicts the results in ref. 8, where bilayer samples do show magnetism;

2. Fig. 1 presents detailed symmetries for one magnon, whereas the Raman signal (at least at zero fields) investigated in the following figures are two-magnon.

3. Page 6, "an additional peak with a central frequency $\sim 7 \text{ cm}^{-1}$...". Obviously, it should be 6 cm^{-1} or even smaller.

Dear Dr. Bladwell:

Thank you for your communication regarding our manuscript NCOMMS-21-37170, titled “Magnons and magnetic fluctuations in atomically thin layers of MnBi_2Te_4 ”. We thank the reviewers for their overall positive evaluations of our work and their thoughtful questions. We have carefully considered their comments and questions. We believe that the revised manuscript is significantly improved. Hopefully, the reviewers agree with us and find the revised manuscript ready for publication in Nature Communications.

Response letter to reviewers

We thank the reviewers for their overall positive evaluations of our work and their thoughtful questions. We believe that the manuscript is significantly improved by addressing the questions and is suitable for publication in Nature Communications.

Before answering the reviewers’ questions in detail, we summarize the major changes made in the revised manuscript:

- We added the temperature-dependent dynamic Raman susceptibility analysis in supplementary information (SI) to address reviewer C’s concern if quasi-elastic scattering as reported reflects spin fluctuations.
- We added thickness-dependent low-frequency phonon spectra in the SI to (i) demonstrate the sample quality of exfoliated thin layers; and (ii) to show spectral overlap between phonons and magnons in layers of different thicknesses.
- We acknowledged that the sample quality in ultra-thin layers is always a concern and we modified our speculation on magnetic order stability in the 1-SL to be more precise.

We now address questions and comments from the reviewers on a point-to-point basis.

Report of Referee A

“In the manuscript “Magnons and magnetic fluctuations in atomically thin MnBi_2Te_4 ” D.Lujan et al. study collective spin excitations (magnons) in ultrathin films of antiferromagnetic topological insulator MnBi_2Te_4 by using Raman spectroscopy technique. They mostly focus on the films composed of 2-seven layer (SL) blocks and study 3 magnetic phases: antiferromagnetic, canted-antiferromagnetic and ferromagnetic ones. After carefully reading the manuscript I clearly see that this is an original paper that contains interesting results on magnetic structure of 2-SL MnBi_2Te_4 system which can be useful for future studies of defect tuning of magnetic properties of MnBi_2Te_4 . The paper is clearly written and can be interesting for wide condensed matter community especially in the field of magnetic topological matter. I can recommend the manuscript for publication in the Nature Communications journal as it is.

Response We thank the reviewer for the positive evaluation and recommendation for publication.

Referee C (Remarks to the Author):

Overall Comments

In the manuscript of D. Lujan et al., Raman spectroscopy was carried out to probe the magnon of low energy and magnetic fluctuation in few-layer magnetic MnBi_2Te_4 . In particular for a 2-SL sample, a signal 5.5 cm^{-1} was assigned to the photon scatterings with two magnons. Its evolution in perpendicular magnetic fields was used to identify various magnetic phases such as AFM, c-AFM and FM. By contrast, this Raman peak was

found to be absent in 1-SL, owing to the strong magnetic fluctuation.

Different from the more commonly used non-contact techniques that characterize magnetization directly, e.g. MCD and MOKE, Raman is used to probe collective magnetic excitations, such as the magnon and quasielastic scattering presented in the manuscript. Although MBT is being intensively studied by various techniques, there has been few papers about Raman characterization of magnons. In this regard, the manuscript is novel and interesting. Nevertheless, the experimental result is not systematic and conclusive. Several issues need to be clarified. ”

Response: We thank the reviewer for the overall positive evaluation of the manuscript. His/her thoughtful questions have motivated us to improve the manuscript further.

Q1: *1-SL has shown to be ferromagnetic at low temperature by MCD and transport measurement. The lack of magnon signal in Raman spectroscopy thus may not be generic. Rather, it could be due to sample degradation.*

In addition, for the thicker samples (> 2-SL), the absence of magnetic signal, even in a strong field that would drive the magnon mode away from the low-energy phonon mode, is surprising. The author should clarify the detailed layer dependence, including both odd and even layers.

Response: We agree with the reviewer that magnon spectra that evolve as a function of layer thickness would be ideal. Unfortunately, we were not able to observe magnons except in the 2-SL sample. We offer several possible reasons. First, the magnon Raman signal is rather weak in MBT in comparison to other vdW magnets, for example, CrI₃. One of us (Prof. Rui He) has had extensive experience with magnon Raman experiments on CrI₃. Secondly, the magnon frequency is rather low, near the detection limit of our high resolution Raman spectrometer. Lastly, few-layer MBT has an ultralow frequency interlayer phonon mode (see Fig. R1 below) whose frequency decreases in thicker layers. The frequency of this ultralow frequency interlayer phonon may overlap with those of magnons in N-layer MBT (with N equal or greater than 3), which would obscure the weak magnon signals in thicker layers.

Figure R1: Layer dependence of low frequency Raman spectra in 1 to 5 SLs. The data is taken at room temperature. The stripe-pattered shade is to block the noise line. The solid lines are Lorentzian fits.

We have speculated that long range magnetic order is less robust in 1-SL MBT in the original manuscript. Our speculation is supported by (i) increased magnetic fluctuations in 1-SL; (ii) the significantly smaller magnetic anisotropy energy in MBT than that in CrI₃, making the magnetic order in 1-SL MBT less robust than that in CrI₃ monolayers; and (iii) the predicted magnon frequency in 1-SL MBT at 6T (spectra in Fig. 2 in the main text) is 6 cm⁻¹ as shown in Fig. R2. This value is within our detection range but not detected.

We understand the reviewer’s concern about sample quality. It is always a concern for ultra-thin vdW samples. The low-frequency Raman spectra shown in Fig. R1 are taken at room temperature using collinearly polarized incident and scattered light. This phonon mode corresponds to a breathing mode. The systematic frequency shift as a function of layer thickness can be modeled by a linear chain model. The mode observed from the 1-SL corresponds to lattice vibrations against the substrate. The presence of this mode suggests that the crystalline structure of 1-SL is retained. We acknowledge that the phonon spectra do not fully address the concern over sample quality but it is an evidence widely used by the community.

Figure R2: Monolayer Magnon Energy at the Gamma point as a function of out-of-plane applied magnetic field.

We also recognize that 1-SL is most likely influenced by coupling to the substrate. We are not aware of many experimental studies down to the 1-SL limit although there are two studies. One study reported MCD signal down to the 1-SL limit [1] although there is a chance that substrate coupling also influenced this MCD measurement since even-layer samples have shown a net moment, which is unexpected. The presence of MCD signal generally means that there is spin imbalance or a net moment. However, it does not mean that a long range magnetic order is present. Another experiment reported transport measurement down to 1-SL in MBE grown sample [2]. However, the resistivity in 1-SL is significantly higher than that measured in 2-SL and higher, leading to the question if there is any significant difference in the 1-SL sample quality.

Revision: We agree with the reviewer that the sample quality and sample-substrate interface quality are important, especially in studies of 1-SL samples. We modified relevant sentences in the main text (marked by red text) and also added the following discussion “We cannot completely rule out the possibility that the absence of magnons in 1-SL is related to coupling to the substrate or reduced sample quality. Future experiments performed under different conditions (e.g. at higher magnetic fields) are needed.”

We also added the following description in the method section. "We present ultralow frequency Raman phonon spectra from flakes with different thicknesses from 1-SL to 5-SL. The systematic frequency shift as well as their reasonably narrow linewidths support the high crystalline quality of these ultrathin layers".

Q2: *As such, a major concern is the quality of samples and the repeatability of experimental results. Encapsulation in BN heterostructures is strongly recommended to obtain reliable signals.*

Response: We are aware that MBT samples can degrade by exposure to ambient conditions. Our samples were prepared in a glove box and were only exposed to air for no more than 1 minutes during the mounting procedure. We have also attempted measuring samples with encapsulation layers. However, the encapsulation layer reduces the polarization contrast in the Raman signal. For this reason, we chose to present data from samples without encapsulation. The fact that we were able to observe magnons on the 2-SL suggests that the sample quality is sufficient for the purpose of our experiments.

Revision: We added the following sentence in the method. "Ultrathin MBT layers may degrade in ambient environment. The samples studied here were prepared in a glove box and exposed to air for no more than 1 min. We have also performed measurements on encapsulated samples but found reduced polarization contrast. Thus, we chose to present results from a sample without encapsulation layers."

Q3: *For the two-magnon mode, any explanation of its narrow width? It is even narrower than the one-magnon mode in high magnetic fields.*

Response: This is an excellent observation that has surprised us as well. Typically, magnon linewidth is related to magnon lifetimes. In insulators, magnon lifetimes are determined by either magnon-magnon scattering or magnon-phonon scattering. The theory presented does not allow us to predict the magnon line width. We also feel that the experimentally extracted linewidth may not be sufficiently accurate to make a reliable statement. For example, the single magnon resonances are only observed at higher magnetic field where there is partial spectral overlap between the phonon and magnon peaks. Thus, we believe that a comparison between the single magnon and two-magnon linewidths is not reliable.

For these practical reasons and limitations, we did not attempt to explain the narrow two-magnon linewidth in MBT in this paper. We have observed narrow two-magnon linewidth in another sample, CoTiO_3 , which is known to host topological magnons in its AFM phase. Our current hypothesis is either hexagonal lattice or magnon topology may be partially responsible for this narrow two-magnon scattering linewidth. We hope that the community can explore this interesting question in future studies.

Revision: We added the following sentence to the main text "In contrast to other AFMs, the narrow linewidth of the two magnon scattering is somewhat surprising and should be further investigated in future studies."

Q4: *Again, why is this mode absent in 4-SL? It's not expected to be obscured by the phonon mode 5 cm^{-1} .*

Response: As the reviewer pointed out, the phonon in the 4-SL was experimentally observed and found to be at 5 cm^{-1} , which is lower than the phonon in the 2-SL at 8 cm^{-1} (see Fig. R1 above). We computed the Gamma point magnons for 4-SL MBT, assuming the same exchange interaction and magnetic ion anisotropy values we obtained for the 2-SL case, defined on the 4-SL structure. We found a Raman-active magnon with energy 5.2 cm^{-1} , which overlaps with the phonon at 0 T. We have searched for a B-field dependent frequency shift as a key signature of the magnon. At 6 T, the calculated magnetic ground state corresponds to a canted phase

with Raman-active magnons at 3.8, 5.7, and 6.4 cm^{-1} , which still have significant overlap with the phonon mode.

Revision: We added the following discussion to the manuscript. “We were not able to observe a systematic evolution in magnon spectra as a function of layer thickness. In 4-SLs, for example, the low-frequency phonon mode spectrally overlaps with and masks the anticipated magnon resonances.”

Q5: *Due to the variation of sample quality, the spin-flop and FM transition fields are sample dependent. The classification of magnetic phase in Fig. 3 seems not reliable.*

(a) Check the data of 3 T in Fig. 3a and Fig. S2. From the raw data, one could not distinguish any sudden change.

Response: While we anticipate a small frequency jump in the magnon frequency from the AFM to c-AFM phase (calculation presented in Fig. 3c in the main text), we agree with the reviewer that the magnon data are not very clear to identify the magnetic field for this transition.

(b) The assignment of 5 T to c-AFM has no evidence, as it could also belong to the FM phase.

Response: For the C-AFM to FM transition, the canted angle decreases as the magnetic field increases until all spins are align and reach the FM phase. Since the transition is smooth, there is no sudden/clear change in the magnon energy from C-AFM to FM transition as shown in the calculation presented in Fig. 3c in the main text.

Overall, we agree with the reviewer that these transitions are difficult to identify only based on magnon measurements. In this paper, we drew the phase diagram and label the transitions based on the reflective magnetic circular dichroism (RMCD) measurements presented in [3] to determine the magnetic fields where these phase transitions occur. We recognize that the exact values can change for different samples. However, both our group and Xu’s group use bulk crystals grown by Dr. Yan (a co-author on both this paper and the published Nano Letter [3]). Thus, we believe that it is reasonable to draw the phase transitions based on published results.

Revision: We added the following discussion. “The exact fields where the transitions from AFM to c-AFM or from c-AFM to FM occur are difficult to determine based on magnon spectra alone. We rely on magnetic circular dichroism spectroscopy measurements performed on similar samples [3] to guide our analysis.”

Q6: *Abnormally strong background signals were found for samples of various thickness at room temperature, which were ascribed to QES.*

(a) Whether is it due to the polymer substrate? The authors need to exclude the fluorescence of organic molecules.

Response: We observed no florescence signal or any other additional Raman signal besides the 521 cm^{-1} peak from silicon Si when we take background measurements on regions without the MBT flakes.

(b) It should be suppressed by magnetic ordering below T_N , at least for 2-SL and thicker sample. The authors should show the critical variation around T_N to prove it is indeed related to magnetic fluctuation.

Response: We thank the reviewer for this suggestion and now address this concern by adding a section in the SI. We analyze temperature dependence of dynamic Raman susceptibility χ''_{dyn} in 2 and 3 SLs (Fig. R3). We observe χ''_{dyn} is peaked at the Nèel temperature, ~ 17 K. The observed Nèel temperature for 2 SLs is consistent with the integrated intensity analysis shown in Fig. 2c. We notice that the difference between 2 and 3 SLs is unclear due to our limited temperature step (2 K). χ''_{dyn} is proportional to magnetic specific heat C_m ,

which shows the critical behavior of magnetic phase transition [4, 5]. This temperature dependence supports our analysis of the QES as evidence for increasing magnetic fluctuations with decreasing layer thickness.

Revision: We added the following discussion to the main text “We observe χ''_{dyn} is peaked at the Néel temperature, ~ 17 K (data in SI), which is expected for the critical behavior of magnetic phase transition [4, 5] and supports our analysis of the QES.”

Figure R3: Temperature dependence of dynamic Raman susceptibility for 2 and 3 SL. The grey shade indicates the critical behavior across the Néel temperature.

Q7: *In the first paragraph, the author said that “... it breaks down in NiPS3 monolayers and bilayers”. This contradicts the results in ref. 8, where bilayer samples do show magnetism.*

Response: Indeed, there were some debates about if the magnetic order is stable in NiPS3 bilayers. To be consistent with the cited reference, we followed the reviewer’s suggestion and revised the sentence to be “...it breaks down in NiPS3 monolayers.”

Q8: *Fig. 1 presents detailed symmetries for one magnon, whereas the Raman signal (at least at zero fields) investigated in the following figures are two-magnon.*

Response: We choose the present one magnon symmetry properties in Fig. 1 intentionally because they are conceptually simpler. However, the reviewer is correct, we need to explain how this magnon symmetry is related to our observation of two magnons scattering in the AFM phase. This contrast in the Raman selection rules between one-magnon and two-magnon scattering processes is one of the evidence that supports our assignment of the two-magnon scattering peak. We added the following sentence in the main text “but in contrast with the selection rule expected for one-magnon scattering (Fig. 1c).”

Q9: *Page 6, “an additional peak with a central frequency ~ 7 cm⁻¹ ...”. Obviously, it should be 6 cm⁻¹ or even smaller.*

Response: We change to the more accurate central frequency of the magnon mode: $\sim 6.2 \text{ cm}^{-1}$.

References

- [1] Yang, S. *et al.* Odd-even layer-number effect and layer-dependent magnetic phase diagrams in mnbi_2te_4 . *Phys. Rev. X* **11**, 011003 (2021). URL <https://link.aps.org/doi/10.1103/PhysRevX.11.011003>.
- [2] Zhao, Y.-F. *et al.* Even-odd layer-dependent anomalous hall effect in topological magnet mnbi_2te_4 thin films. *Nano letters* **21**, 7691–7698 (2021).
- [3] Ovchinnikov, D. *et al.* Intertwined topological and magnetic orders in atomically thin chern insulator mnbi_2te_4 . *Nano Letters* **21**, 2544–2550 (2021). URL <https://doi.org/10.1021/acs.nanolett.0c05117>.
- [4] Reiter, G. F. Light scattering from energy fluctuations in magnetic insulators. *Phys. Rev. B* **13**, 169–173 (1976). URL <https://link.aps.org/doi/10.1103/PhysRevB.13.169>.
- [5] Kim, K. *et al.* Suppression of magnetic ordering in xxz-type antiferromagnetic monolayer nips 3. *Nature communications* **10**, 1–9 (2019).

Reviewers' Comments:

Reviewer #3:

Remarks to the Author:

The present version has addressed most of my concerns. I recommend its publication in Nature Communications.